# Uterine Factor Infertility, a Systematic Review

**DOI:** 10.3390/jcm11164907

**Published:** 2022-08-21

**Authors:** Camille Sallée, François Margueritte, Pierre Marquet, Pascal Piver, Yves Aubard, Vincent Lavoué, Ludivine Dion, Tristan Gauthier

**Affiliations:** 1Department of Gynecology and Obstetrics, Mother and Child Hospital, University Hospital Center of Limoges, 87000 Limoges, France; 2Department of Gynecology and Obstetrics, Intercommunal Hospital Center of Poissy-Saint-Germain-en-Laye, 78103 Poissy, France; 3Department of Pharmacology and Toxicology, Centre Hospitalier Universitaire de Limoges, 87042 Limoges, France; 4Department of Obstetrics and Gynecology, Hopital Universitaire de Rennes, 35000 Rennes, France

**Keywords:** uterine factor infertility, absolute uterine factor infertility, non-absolute uterine factor infertility, congenital uterine dysfunction, acquired uterine dysfunction

## Abstract

Uterine factor infertility (UFI) is defined as a condition resulting from either a complete lack of a uterus or a non-functioning uterus due to many causes. The exact prevalence of UFI is currently unknown, while treatments to achieve pregnancy are very limited. To evaluate the prevalence of this condition within its different causes, we carried out a worldwide systematic review on UFI. We performed research on the prevalence of UFI and its various causes throughout the world, according to the PRISMA criteria. A total of 188 studies were included in qualitative synthesis. UFI accounted for 2.1 to 16.7% of the causes of female infertility. We tried to evaluate the proportion of the different causes of UFI: uterine agenesia, hysterectomies, uterine malformations, uterine irradiation, adenomyosis, synechiae and Asherman syndrome, uterine myomas and uterine polyps. However, the data available in countries and studies were highly heterogenous. This present systematic review underlines the lack of a consensual definition of UFI. A national register of patients with UFI based on a consensual definition of Absolute Uterine Factor Infertility and Non-Absolute Uterine Factor Infertility would be helpful for women, whose desire for pregnancy has reached a dead end.

## 1. Introduction

Infertility affects an average of about one in five couples, representing a real public health problem [1]. It can be primary or secondary, in about one third and two thirds of cases, respectively [2,3]. It may be of female and/or male origin, or even of undetermined cause in about 10% of cases [4].

Uterine factor infertility (UFI) is defined as a complete lack of a uterus (Absolute Uterine Factor Infertility or AUFI) or as a nonfunctional uterus (Non-Absolute Uterine Factor Infertility or NAUFI). The exact prevalence of UFI is currently unknown. Early studies (1970s), which have been repeatedly conducted over the years, suggest that it affects 3–5% of the world’s female population and that AUFI affects up to 1 in 500 women of childbearing age [5,6].

There are many causes of UFI, congenital and acquired [7,8]: uterine agenesis, hysterectomies, uterine malformations, polyps, myomas, adenomyosis, synechiae, uterine irradiation.

To our knowledge, there is no reliable data of the prevalence of UFI and its various causes among women under 40 years old.

For women who do not have a uterus, treatments to achieve pregnancy are highly limited. Uterus transplants, which are still at a stage of research, represent a real hope for these patients [9,10].

To have a better knowledge of the potential needs for uterine transplants or even surrogacy, epidemiological data on women of childbearing age are needed.

Therefore, to evaluate the prevalence of UFI, we carried out a systematic review on UFI, aiming to improve and adapt treatments and public health policies to the needs of the population suffering from this condition.

## 2. Materials and Methods

For this systematic review, we wanted to get the lay-of-the-land of the prevalence of UFI and its various causes throughout the world.

We discussed AUFIs:-Congenital: uterine agenesis (MRKH syndrome, complete androgen insensitivity syndrome (CAIS));-Acquired: hemostasis hysterectomy (postpartum hysterectomy), hysterectomy for benign conditions (polymyomatous uterus, endometriosis, adenomyosis, functional menorrhagia, pelvic statics disorder), carcinological hysterectomy (ovarian, endometrial or cervical cancer).

We also studied NAUFIs:

-Congenital: uterine malformations, Distilbene^®^ uterus;-Acquired: uterine irradiation, adenomyosis, synechiae and Asherman’s syndrome, myomas, polyps.

We focused on data concerning women of childbearing age, under 40 years.

We registered our study in the PROSPERO database under the number CRD42021254994.

According to the PRISMA criteria [11], we performed several searches using the PubMed search engine for publications containing the following keywords:-“uterine factor infertility”;-“uterus transplantation”;-“adenomyosis [and] infertility”;-“uterus agenesis [and/or] MRKH syndroma [and/or] complete androgen insensitivity syndroma [and] infertility”;-“hysterectomy [and] prevalence [and/or] incidence”;-“uterus malformation [and/or] hypoplastic uterus [and/or] uterus septa [and/or] DES uterus [and] infertility”;-“uterine myomas [and] infertility”;-“uterine polyp [and] infertility”;-“uterine synechia [and] infertility”;-“uterus radiation [and] infertility”.

Trying to obtain up-to-date results, we limited our search to material published between 2000 and 2021. After removing duplicates, we excluded texts whose title or abstract did not include the above keywords. We also excluded case reports, as they did not allow us to obtain valuable epidemiological results. Letter-like articles were also excluded for the same purpose. We only selected publications written in English or French. This work was carried out using Excel^®^ software (version 16.64, 2022, Microsoft^®^, Redmond, WA, USA).

All of the publications initially selected were red by two obstetrician-gynecologists who kept those that met the objective.

The inclusion criteria for this systematic review were prospective, retrospective, and cross-sectional studies that assessed the prevalence of the various uterine conditions mentioned above, particularly in women under 40 years old, and their impact on fertility. Papers with data on incidence, but none on prevalence, were included in the study with a specific mention of this feature. Quality assessment of the selected studies was performed using the New-Castel Ottawa Scale [12].

## 3. Results

The following flow chart (Figure 1) was designed according to PRISMA criteria. Out of 477 articles reviewed for eligibility, only 188 were epidemiological studies that specifically evaluated prevalence and/or incidence data for UFI. The 289 remaining studies not included in the qualitative synthesis mentioned these data, but only in the background or they had been obtained from other systematic reviews.

### 3.1. Uterine Factor Infertility: General Data

We found four studies that assessed the prevalence of uterine factor infertility in their respective countries [3,13,14,15]. Details of these studies are summarized in Table 1.

Out of the four studies, only one of them was conducted prospectively and provided overall results on infertility within the country.

In the study by Meng et al., which is a prospective study of Chinese couples of childbearing age who attended premarital counseling, infertility affected 13.6% of the couples with 14.0% primary infertility and 11.2% secondary infertility among the Chinese general population [15]. The cause was male in 17.0% of cases, female in 40.0%, mixed in 26.0% and unexplained in 17.0%. UFI accounted for 12.1% of female infertility cases [15].

The other three epidemiological studies involved couples attending assisted reproduction centers for infertility of at least 12-month duration. Uterine factor infertility accounted for 2.1 to 16.7% of the causes of female infertility [3,13,14].

According to French data published by INSERM (National Institute of Health and Medical Research) in 2018, 10% of couples are infertile in France, without any details on the causes of infertility [1].

### 3.2. Uterine Agenesia

Only one epidemiological study had assessed the prevalence of MRKH syndrome. This was a Danish retrospective study from 2016, conducted from 1994 to 2015. The study used the Danish register of public hospital inpatients and the Danish Central Cytogenetic Register. The prevalence was estimated by considering women born between 1974 and 1996. All potential diagnoses of MRKH syndrome were checked in the patients’ medical records. In this study by Herlin et al., with 138 women born with medically confirmed MRKH and 687,517 female births over the same period, the prevalence of MRKH syndrome among female birth was 1/4982 (95% CI: 4216–5887), or 0.02% [16].

So far, there is no calculation of the prevalence of CAIS drawn from epidemiological studies nor general population estimates [17]. However, the minimal incidence of Androgen Insensitivity Syndrome is estimated at 1/99,000 based on patients with molecular proof of the diagnosis in the Netherlands [18]. Among girls with inguinal hernias, the prevalence of CAIS is estimated between 0.8% and 2.4% [19]. 

### 3.3. Hysterectomies

A consequent amount of data on hysterectomies was available but highly heterogeneous.

The annual incidence of hysterectomy ranges from 70 women per 100,000 per year in Australia to 700 per 100,000 in the USA [20,21,22,23,24,25,26,27,28] (Table 2). Two studies assessed this incidence from samples drawn from national patient cohort follow-ups, comprising randomly selected people from the general population. Questionnaires had been sent to women from these cohorts asking whether they had had a hysterectomy and at what age they had been operated [20,21]. The other studies used existing registers from these countries and standardized the rates based on the age of the national population [22,23,24,25,26,27,28].

The prevalence of hysterectomy among women under 40 ranged from 1.7% in India to 14.0% in the United States [29,30,31,32,33,34,35,36,37,38,39,40,41,42], the former using a questionnaire and the latter a register (Table 3). For five studies, data had been obtained from questionnaires sent to a randomly selected national cohort of patients [31,38,40,41,42]. For seven studies, data had been collected from national registers with age standardization [29,32,34,35,36,37,39]. Finally, two studies using patient cohorts had assessed the prevalence of hysterectomy in women of childbearing age: the first one had assessed the prevalence by sending a questionnaire and had checked each positive answer to a history of hysterectomy with an ultrasound during a dedicated consultation; the second consisted of creating a cohort of patients, consulting the gynecologists participating in the study for their follow-up and whose files were studied in the event of a hysterectomy [30,33].

In Taiwan, in a retrospective study of women with insurance coverage, conducted in 2010, 12.1% of hysterectomies were performed before the age of 40 years [43].

In the United States, according to Merrill et al., in a retrospective study based on Utah registers and using age-adjustment, the main indications for the 2910 hysterectomies performed on patients aged 25–34 were functional metrorrhagia (23.0%), endometriosis (17.3%), prolapse (15.7%), gynecological cancers (5.4%) and myomas (4.1%) [26]. In contrast, in China, according to a retrospective study based on 4653 hysterectomies with age stratification, the main indications in 110 women aged 20–29 years were gynecological cancers (57.7%), hemostasis hysterectomies (10.6%) and adenomyosis (6.5%) [44].

Regarding hysterectomies due to benign conditions, we found two epidemiological studies that described their annual incidence in women of childbearing age [45,46]. Parazzini et al. found an annual incidence of 200 cases per 100,000 persons a year [46]. This was a retrospective study of all hysterectomies performed in Lombardy from 1991 to 2016, with a total of 143,045 hysterectomies. The annual incidence by age group was estimated by taking the yearly number of hysterectomies performed on patients aged 20–34 as the numerator and the female population of Lombardy of the same age as the denominator, obtained from a local register [46].

Chen’s study, retrieved from the Canadian Institute for Health Information Discharge Abstract Database, listed all diagnostic and procedure codes for all treated inpatients, as well as the number of hysterectomies performed in 2007 in Ontario (13,511) and calculated age-standardized rates for women living in Ontario. The annual incidence in this study was 260 cases per 100,000 persons a year in women of childbearing age [45].

We considered emergency hysterectomies in case of obstetrical complications (post-partum hemorrhage, placenta accreta, placenta previa) as hemostasis hysterectomies.

We identified 102 epidemiological studies that evaluated the annual incidence of hemostasis hysterectomies among deliveries by country. We categorized them by continent for clarity.

For the African continent, according to 17 epidemiological studies, the annual incidence of hemostasis hysterectomies among all deliveries ranged from 0.12% in South Africa to 1.25% in Niger [47,48,49,50,51,52,53,54,55,56,57,58,59,60,61,62,63].

For the American continent, we obtained data from 11 studies encompassing only the USA and Canada. The annual incidence in these countries ranged from 0.05% in Canada to 0.27% in the USA [64,65,66,67,68,69,70,71,72,73,74].

For the Asian continent, according to 36 studies, the annual incidence ranged from 0.01% in Japan to 0.69% in India [75,76,77,78,79,80,81,82,83,84,85,86,87,88,89,90,91,92,93,94,95,96,97,98,99,100,101,102,103,104,105,106,107,108,109,110].

For the European continent, according to 32 studies, it ranged from 0.02% in Norway to 0.22% in Italy [111,112,113,114,115,116,117,118,119,120,121,122,123,124,125,126,127,128,129,130,131,132,133,134,135,136,137,138,139,140,141,142].

Finally, for Oceania, we found one study from New Zealand and five from Australia. In these countries, the annual incidence of hemostasis hysterectomy ranged from 0.04 to 0.12% [143,144,145,146,147,148].

A study from 2020 encompassing nine European countries estimated the incidence of hemostasis hysterectomy in 2012–2013 at 0.06% [124]. However, data are still evolving as post-partum hemorrhage management has recently improved with the use of embolization or Bakri balloon, even though evidence in reducing hemostasis hysterectomy is still being debated [149].

We considered hysterectomies performed for endometrial, cervical and ovarian cancer on women under 40 years of age as carcinologic hysterectomies. Related data for this purpose was scarce in the literature.

We only found three epidemiological studies that looked at carcinologic hysterectomies in the USA.

Garg et al. conducted a retrospective study of endometrial cancers in women under 40 years of age. Out of 2000 hysterectomies performed for endometrial cancer during the study from 1993 to 2008, 70 patients were under 40 years of age (3.5%), with an average age of 37 years and ages ranging from 24 to 40 years [150]. Similarly, in this study, 23% of patients under 40 years of age reported infertility prior to their hysterectomy and 76% were nulliparous [150].

In a cross-sectional study conducted in 2012, Esselen et al. estimated that 46,450 hysterectomies were yearly performed in the United States for gynecological cancers: 61% for endometrial cancer, 9% for cervical cancer, 27% for ovarian cancer and 3% for gynecological cancers of undetermined origin [151]. Among the 28,160 cases of endometrial cancer, 2.7% of patients who had a hysterectomy were between 18 and 39 years of age [151]. Among the 4045 cases of cervical cancer, 30.2% of patients who had a hysterectomy were between 18 and 39 years of age [151].

In a retrospective study conducted by Merril et al., from 1998 to 2002, out of 45,784 hysterectomies performed over that period, 51.1% were due to endometrial cancer, 29.8% to ovarian cancer and 19.1% to cervical cancer [152].

### 3.4. Uterine Malformations

To define uterine malformations, we referred to the ESHRE/ESGE classification from 2014 [153].

According to our research, three epidemiological studies focused on congenital uterine malformations without making any distinction between them.

According to a 2002 retrospective Chinese study, the prevalence of infertility in patients with any type of uterine malformation was 26.6% [154].

Two studies assessed the prevalence of uterine malformations among infertile female populations. A prospective Canadian study evaluating the performance of hysterosonography in infertility or metrorrhagia estimated the prevalence at 20.0%, with 15.0% arched uterus (U1c), 4.6% uterine septum (U2a and b), 0.2% unicornuate uterus (U4b) and 0.2% hypoplastic or infantile uterus (U1b) [155]. In a prospective English study of patients consulting for infertility or subfertility, the prevalence of uterine malformations in this population was 28.2%, with 16.3% arched uterus (U1c), 1.2% uterine septum (U2a and b), 0.5% unicornuate uterus (U4b), 0.2% bicornuate uterus (U3a) and 0.04% didelphic uterus (U3bC2) [156].

In addition, four studies focused on uterine septa, representing the most common major uterine malformations (arched uteri are considered as a minor malformation). Ludwin et al., in a prospective study, found up to 35% septum cases in the infertile population [157]. According to two retrospective studies, the prevalence of infertility in the case of uterine septum was 54.5 to 55.1% [158,159]. Finally, the study by Wang et al. estimated that the prevalence of infertility depended on the size of the septum with an increase from 35.4 to 45.7% [160].

No studies were found regarding the prevalence of infertility for dysmorphic uteri (U1a).

### 3.5. Radiation-Induced Uterine Condition

Uteri that have become hypoplastic as a result of pelvic irradiation were considered as, radiation-induced uterine condition

To our knowledge, there are no data about the prevalence of radiation-induced uterine condition at the international or national level. According to two Danish and French retrospective studies, the prevalence of infertility in cases of direct uterine or pelvic irradiation is 81–81.3% [161,162]. However, according to these two studies, the consequences of radiotherapy on the ovaries also played a role in this high prevalence of infertility.

### 3.6. Adenomyosis

We found three epidemiological studies that evaluated the prevalence of adenomyosis and its impact on fertility.

In a German prospective study from 2005, the prevalence of adenomyosis, diagnosed on MRI, in non-menopausal women was 28% and reached 79% in the population with endometriosis [163].

Furthermore, according to a French cross-sectional study from 2020, which included women between 18 and 42 years old who had been surgically explored for benign gynecological conditions and had had Magnetic Radiological Imaging (MRI) performed by a senior radiologist, the prevalence of infertility was 30.2% among women with all types of adenomyosis. Furthermore, primary infertility was evaluated at 19.8% and secondary infertility at 10.4% among women with adenomyosis [164].

Finally, in an Egyptian cross-sectional study from 2020, the prevalence of adenomyosis assessed by ultrasound was 7.5% in an infertile population under 41 years of age [165].

### 3.7. Synechiae and Asherman Syndrome

We did not find any epidemiological study evaluating the prevalence of synechiae in the general population.

Salzani et al., in a cross-sectional study, estimated the prevalence of synechiae at 37.6% in the case of a recent history of curettage for a pregnancy terminated before 20 weeks of gestation [166].

Concerning the prevalence of infertility in case of synechiae, three epidemiological studies reported results ranging from 41.0% in Romania to 90.8% in France, and 60.4% in Tunisia, keeping in mind that in the French study, most patients presented with Asherman syndrome with severe synechiae [167,168,169].

### 3.8. Uterine Myomas

To our knowledge, only one US prospective study, conducted in 2002 on 3000 pregnant women, assessed the prevalence of uterine myomas in non-infertile women of childbearing age. This prevalence was 11% [170].

Regarding fertility data, we found four epidemiological studies [171,172,173,174] with very different results that are synthesized in Table 4.

In the Italian study, patients had other causes of infertility other than myomas [171]. In the Indian study, only the prevalence of infertility for submucosal myomas was estimated [174].

We documented two French prospective studies by the same author that evaluated the prevalence of infertility in myomas before and after embolization. Before treatment, the prevalence of infertility was estimated at 32.3–33.3% in the presence of multiple uterine myomas (more than 3) [175,176].

### 3.9. Uterine Polyps

We did not find any epidemiological study on the prevalence of uterine polyps in the general population or on the prevalence of infertility in cases of uterine polyps.

According to the only epidemiological study we found in the literature, the prevalence of polyps in the Egyptian infertile population was 15.6% [177].

## 4. Discussion

This systematic review shows the vast heterogeneity and lack of data concerning UFI in many countries. It also shows the complexity of this multifactorial condition, which could be confusing for the diagnosis.

Preliminary studies on uterine transplantation are based on old studies stating that uterine factor infertility affects 3 to 5% of the world’s female population and absolute uterine factor infertility affects 1 in 500 women of childbearing age [5,6,178,179,180]. However, we cannot confirm these data. One of our issues is that the definitions of UFI differ from one study to another. For example, we considered uterine irradiation, Asherman syndrome and DES syndrome as NAUFI, whereas some studies considered them as a cause of AUFI [180,181]. In their severe form, these uteri become non-functional and for patients who wish to carry a pregnancy, no treatment other than a possible uterine transplantation is available. In fact, a simple definition of NAUFI is needed to clearly estimate the prevalence of this condition, and the needs among the general population: this could be based on the presence of a pathological uterus (clinical or imaging-based examinations) associated with either over-one-year-long infertility or failure of embryo transfer.

Moreover, the studies did not necessarily use the same denominator to give the percentages. Therefore, some elements of data were difficult to compare. This is the case for the study by Meng et al. [15] which did not give the proportion of primary and secondary infertility in the infertile population but the data for primary and secondary infertility in the general population (see Table 1). Furthermore, incidence and prevalence were often mixed up in many studies. We also had to face problems concerning some diseases such as myomas, uterine irradiation, synechiae and uterine malformations which do not have the same impact on fertility depending on their severity, location and/or size. 

The heterogeneity of the data was also created by diverse factors: the proportion of causes of UFI varied, samples which could not be compared, and some specific data were not always available in several countries.

However, we have observed that UFI represents a significant proportion of female infertility with a rate ranging from 2.1 to 16.7%, depending on the study [14,15]. Consequently, an accurate and consensual definition of absolute and non-absolute uterine factor infertility among the different pathological contexts identified (congenital anomaly, adenomyosis, myoma…) is urgently needed. It will allow researchers to carry out incidence studies and to develop research into dedicated therapies.

One of the strengths of our study is that it looked at all uterine conditions that may cause UFI, both absolute and non-absolute, for which we provide accurate definitions. It is also the only review that has favored epidemiological studies, to find out the prevalence of UFI and its different causes.

Furthermore, our search was comprehensive due to the large number of articles studied.

Finally, we followed the PRISMA criteria and assessed the quality of all the selected epidemiological studies we selected using the validated Newcastle-Ottawa Scale. Our study is also registered on the PROSPERO website.

The vast heterogeneity of the studies did not allow us to estimate the precise prevalence of many conditions (myomas, adenomyosis, hysterectomies before age 40, polyps) and their impact on fertility, which is one of the weaknesses of our study. This heterogeneity prevented us from carrying out a meta-analysis either and led to an evaluation bias. Moreover, we took account of the prevalence of the diseases but sometimes we did not know if the patients still had a desire for pregnancy (MRKH syndrome, hysterectomies, radiation-induced uterine conditions, …).

## 5. Conclusions

This systematic review enabled us to assess the current state of UFI on a global and national scale and to highlight the lack of studies in the field.

However, it would be interesting to have a national register of patients with uterine factor infertility based on a consensual definition of AUFI and NAUFI. Such a register might allow us to have a better understanding of patients for whom the possibility of carrying a pregnancy is complicated. It could improve their follow-up and help in finding with them a solution adapted to their condition: facilitated adoption, participation in research protocols in uterus transplantation and information about surrogacy.

Concerning uterus transplants, this register may allow us to make a better assessment of the need for this procedure within a country. We would know the number of territorial transplant centers that need to be opened and the costs that this would generate for society.

## Figures and Tables

**Figure 1 jcm-11-04907-f001:**
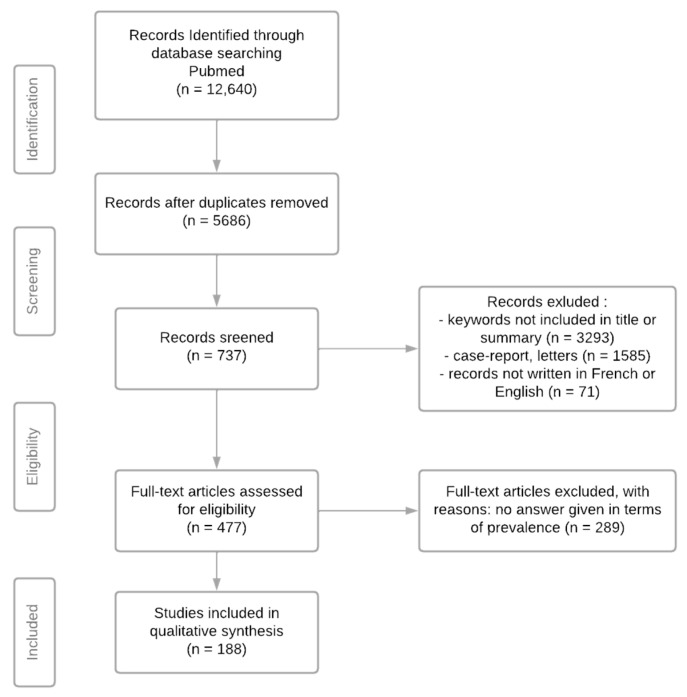
Flow chart.

**Table 1 jcm-11-04907-t001:** Epidemiological data on couple infertility.

Author	Publication Date	Population	Country	Couple’s Infertility (%)	Primary Infertility (%)	Secondary Infertility (%)	Male Infertility * (%)	Female Infertility * (%)	Mixed Infertility * (%)	Unexplained Infertility * (%)	Uterine Factor Infertility * (%)
Benbella et al. [13]	2018	Infertile population (1265 couples)	Morocco		77.2	22.8	28.2	39.6	17.0	15.2	12.6
Elhussein et al. [3]	2019	Infertile population (800 couples)	Sudan		68.9	31.1	35.5	42.8	18.4	3.4	2.1
Masoumi et al. [14]	2015	Infertile population (1200 couples)	Iran		69.5	30.5	66.0	88.9			16.7
Meng et al. [15]	2015	General population (2151 couples)	China	13.6	14.0	11.2	17.0	40.0	26.0	17.0	12.1

* Prevalence among Infertile women or men or couples.

**Table 2 jcm-11-04907-t002:** Annual incidence of hysterectomies by country in women under 40.

Author	Publication Date	Study Design	Country	Annual Incidence (Cases/100,000 Persons a Year)
Babalola et al. [25]	2007	Retrospective study	USA	430
Cooper et al. [20]	2005	Cross-sectional study	UK	150
Desai et al. [21]	2017	Cross-sectional study	India	100
Hammer et al. [22]	2019	Retrospective study	Denmark	150
Hammer et al. [23]	2017	Retrospective study	Denmark	90
Hill et al. [24]	2010	Cross-sectional study	Australia	150
Merrill et al. [26]	2001	Cross-sectional study	USA	700
Redburn et al. [27]	2001	Retrospective study	UK	350
Wilson et al. [28]	2017	Retrospective study	Australia	70

**Table 3 jcm-11-04907-t003:** Prevalence of hysterectomies by country in women under 40.

Author	Publication Date	Study Design	Country	Prevalence (%)
Beckmann et al. [29]	2003	Retrospective study	Australia	7.90
Bower et al. [30]	2009	Retrospective study	USA	4.00
Desai et al. [31]	2019	Cross-sectional study	India	3.59
Gartner et al. [32]	2020	Retrospective study	USA	6.00
Liu et al. [33]	2017	Cross-sectional study	China	3.32
Merrill et al. [34]	2008	Retrospective study	USA	10.00
Merrill et al. [35]	2008	Retrospective study	USA	14.00
Meher et al. [40]	2020	Retrospective study	India	4.10
Meher et al. [41]	2020	Retrospective study	India	3.20
Prusty et al. [42]	2018	Cross-sectional study	India	1.70
Rositch et al. [36]	2014	Retrospective study	USA	10.00
Ruiz de Azua Unzurrunzaga et al. [37]	2019	Retrospective study	Scotland	10.00
Shekhar et al. [38]	2019	Cross-sectional study	India	4.80
Temkin et al. [39]	2018	Retrospective study	USA	10.00

**Table 4 jcm-11-04907-t004:** Prevalence of infertility in presence of uterine myomas.

Author	Publication Date	Study Design	Country	Infertility’s Prevalence (%)
Di Gregorio et al. [171]	2002	Retrospective study	Italy	70.00
Nicolaus et al. [172]	2019	Retrospective study	USA	15.70
Rovio et al. [173]	2012	Retrospective study	Finland	12.00
Roy et al. [174]	2010	Retrospective study	India	44.00

## Data Availability

Not applicable.

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
