# Peer review of "Uterine Factor Infertility, a Systematic Review"

_jcm, 2022, doi:10.3390/jcm11164907_

Round 1

Reviewer 1 Report

The topic of your work is "Uterine Infertility", it is a mistake to include a lack of uterus, when we deal with infertility we manage all the pathologies, structural and/or functional of the uterus that may affect the fertility and the management of those. I understand that to talk about the prevalence of hysterectomies with an analysis per country is something to include in a different work and not in an uterine infertility review. Hysterectomy performed in women under 40 y/o does not mean that there is still a future childbearing desire, and young women without uterus but with no desire for more children do not have a fertility problem.

In your methodology the authors include the "DES" for bibliography review (- “uterus malformation [and/or] hypoplastic uterus [and/or] uterus septa [and/or] 78 DES uterus [and] infertility”) but there is no mention to the dysmorphic uterus (Class U1a of ESGE/ESHRE classification).

If the authors wish to focus on uterine infertility it is important to exclude hysterectomy and focus on the pathology that affect fertility and we manage in our practice in the fertility units.

Author Response

Thank you very much for your comments.

We knew that there was a bias in taking the prevalence of hysterectomies.

However, and we found this for all conditions, we do not know in any case whether patients want to become pregnant. Not all patients with MRKH syndrome desire pregnancy either. These are potential uterine factor infertilities.

Moreover, some patients under 40 who had a hysterectomy still have a desire for pregnancy. That is why we decided to include them and only take the prevalence of disease.

To justify this, we added a sentence to the discussion.

Concerning the uterus malformation, we considered all the malformations and the dysmorphic one too. We based our research according to the ESGE/ESHRE classification.

We hope that we have answered your comments as well as possible.

Yours sincerely

Reviewer 2 Report

This is a systematic review of uterine infertility. This review systematically analyzes the English and French written literature on the subject. All types and causes are discusses separately including its incidence and prevalence. As authors concede, the literature sources are burdened with great heterogenity of data that limits a perfect statystical analysis. Methodology of this study is well-done. The authors followed the PRISMA guidelines and registered the trial at PROSPERO.

The authors conclude the following points:

1. there is heterogenity among studies and countries including the basic terminology;

2. a national register of patients with uterine infertility may provide better treatment and follow-up of this cohort of patients.

Apart from some typos and minor English mistakes, there are no significant objections to this study from my side. I suggest the paper is subjected to proofreading and English language editing.

I congratulate the authors to the great amount of work done and put together in the form of this well-written paper.

Author Response

Thank you very much for your comments.

We sent the manuscript to a translator and now, you have a reread version.

Round 2

Reviewer 1 Report

Still no mention of the ESHRE/ESGE Uterus U1a (dysmorphic)

The rest was corrected

Author Response

Thank you for your comments.

We did not find any studies about U1a specifically. We had a sentence about this.